# Evaluation of Nutritional Components, Phenolic Composition, and Antioxidant Capacity of Highland Barley with Different Grain Colors on the Qinghai Tibet Plateau

**DOI:** 10.3390/foods11142025

**Published:** 2022-07-08

**Authors:** Bin Dang, Wen-Gang Zhang, Jie Zhang, Xi-Juan Yang, Huai-De Xu

**Affiliations:** 1College of Food Science and Engineering, Northwest A&F University, Yangling 712100, China; 2008990019@qhu.edu.cn; 2Laboratory for Research and Utilization of Qinghai Tibet Plateau Germplasm Resources, Qinghai Tibetan Plateau Key Laboratory of Agricultural Product Processing, Academy of Agriculture and Forestry Sciences, Xining 810016, China; 2017990098@qhu.edu.cn (W.-G.Z.); 2015990070@qhu.edu.cn (J.Z.); 3Academy of Agriculture and Forestry Sciences, Qinghai University, Xining 810016, China

**Keywords:** colored highland barley, varieties, nutritional quality, phenolic profile, antioxidant activity

## Abstract

The nutritional composition, polyphenol and anthocyanin composition, and antioxidant capacity of 52 colored highland barley were evaluated. The results showed that the protein content of highland barley in the black group was the highest, the total starch and fat contents in the blue group were the highest, the amylose content in the purple group was quite high, the fiber content in the yellow group was quite high, and the β-glucan content of the dark highland barley (purple, blue and black) was quite high. The polyphenol content and its antioxidant capacity in the black group were the highest, while the anthocyanin content and its antioxidant capacity in the purple highland barley were the highest. Ten types of monomeric phenolic substances were the main contributors to DPPH, ABTS, and FRAP antioxidant capacity. All varieties could be divided into four categories according to nutrition or function. The grain color could not be used as an absolute index to evaluate the quality of highland barley, and the important influence of variety on the quality of highland barley also needed to be considered. In actual production, suitable raw materials must be selected according to the processing purpose and variety characteristics of highland barley.

## 1. Introduction

Highland barley (*Hordeumvulgare* L. *var. nudum Hook. f*.) belongs to the family Gramineae and is a variant of barley. Due to the separation of inner and outer glumes and bare kernels, it is also known as naked barley [1,2]. Highland barley is the main crop and main staple food of farmers and herdsmen on the Qinghai Tibet Plateau. It has the nutritional composition of “three highs and two lows” (namely, high in protein, fiber, and vitamins and low in fat and sugar) [3]. The β-glucan content of highland barley, which is widely distributed in the cell wall and aleurone layer of endosperm, is as high as 3.66–8.62% and has the effects of reducing blood lipid, blood glucose, antitumor, and anti-cardiovascular disease and of improving immunity [4,5]. Highland barley is rich in phenolic compounds (2.42–7.33%), including phenolic acids, flavonoids, and anthocyanins in bound or free states, and has strong free radical scavenging capacity. It is one of the important natural antioxidants, which can regulate human glucose and lipid metabolism and prevent and treat metabolic diseases [2,6,7]. Epidemiological studies have shown that the long-term consumption of highland barley can prevent hyperlipidemia, diabetes, and atherosclerosis [8]. Therefore, highland barley, as one of the high-quality grain raw materials for developing functional food [9], is favored by many people. At present, highland barley has begun to be transferred from production areas to non-production areas and has developed from a regional ration crop to a global healthy food source crop. However, China is rich in highland barley varieties, with thousands of varieties in different colors and shapes, which have led to blindness and lack of follow-up in the consumption, development, and utilization of highland barley along with a lack of scientific basis and guidance. Therefore, how to evaluate and screen high-quality highland barley resources is the key to consuming highland barley.

Highland barley is divided according to grain color, and there are mainly black, blue, purple, and yellow color types in production [2]. Previous studies have shown that the nutritional and functional quality of colored barley is generally higher than that of ordinary barley (white barley). Because it is rich in β-glucan and phenols, colored barley has become a more precious cereal resource and thus has attracted extensive attention [10,11,12]. Lin et al. reported the content of bioactive components in black, blue, and white highland barley and showed that the average contents of total phenols, total flavonoids, procyanidins, anthocyanins, and β-glucan, as well as the phenolic antioxidant capacity in black highland barley are the highest, followed by blue and white highland barley. Phenolic compounds are significantly related to antioxidant capacity [13]. Suriano et al. confirmed through study that among 20 Italian colored genotype barley varieties, the contents of protein and β-glucan of blue barley were the highest. Moreover, ferulic acid was the main bound phenolic acid, salicylic acid and gallic acid were the main free phenolic acids, and the total phenolic content was positively correlated with the antioxidant capacity. At the same time, it was also found that there were significant differences in the respective contents of anthocyanins, carotenoids, and tocopherol of different barley varieties [1]. Ge et al. also confirmed that colored highland barley (white, yellow, black, and blue) is rich in phenolic compounds and that the color of highland barley is different, as are its phenolic acids, flavonoids, and antioxidant capacity [8]. At the same time, recent studies have also pointed out that the nutritional quality, polyphenol content, and antioxidant capacity of barley are comprehensively affected by variety, growth location, environmental conditions, and growth year [14,15].

In recent years, some reports have been made on the nutrients, active components, and functions of highland barley [16,17,18]. However, the main problem lies in that the differences in main nutrients and phenolics contained in highland barley with different varieties and grain colors in large group samples are generally not considered, and this in turn leads to failure in fully reflecting the potential value of highland barley with different grain colors. Consequently, it is difficult to classify, evaluate, and screen high-quality highland barley resources in actual production, and this is harmful to the food industrial application and healthy consumption of highland barley resources [2,13,19]. At the same time, the evaluation of highland barley with different colors mainly focuses on β-glucan and phenolic compounds while ignoring the role of their nutrients in their quality evaluation, thus resulting in insufficient scientific basis for the quality evaluation and consumption choice of highland barley with different grain colors. In view of this, the present study selected 52 varieties of highland barley with different grain colors on the Qinghai Tibet Plateau and compared and analyzed the nutritional components, phenolic content, composition, and antioxidant capacity of highland barley with different grain colors from a group-based perspective. We aimed to determine the quality characteristics and differences in highland barley with various grain colors, to clarify the relationship between main nutrients and phenolic content, and to determine the impact of main phenolic compounds on antioxidant capacity. This will aid us in more scientifically evaluating the quality characteristics of highland barley with different grain colors on the Qinghai Tibet Plateau. The results of this study provide a theoretical basis for breeding colored highland barley varieties with outstanding characteristics, improving people’s understanding of colored highland barley, guiding the healthy consumption of colored highland barley, and promoting high-value development and utilization.

## 2. Materials and Methods

### 2.1. Chemicals and Materials

See Table 1 for the grain colors and variety names of the highland barley covered in this study. All highland barley varieties tested were provided by the Qinghai Academy of Agricultural and Forestry Sciences. The 52 types of highland barley selected in the test were divided into four groups according to grain color—namely purple, blue, black, and yellow—which included 14, 12, 14, and 12 varieties, respectively. All of the tested materials were planted in 2019 in the experimental field of the Qinghai Academy of Agricultural and Forestry Sciences (Xining, Qinghai) (36°67′ N 101°77′ E, altitude 2300 m) under the same soil fertility, cultivation, and climate conditions.

The total starch, amylose/amylopectin, and mixed-linkage beta-glucan kits were provided by Megazyme Co. (Wicklow, Ireland). The 1,1-Diphenyl-2-picrylhydrazylradical (DPPH), 2,4,6-tripyridyl-s-triazine (TPTZ), 2,20-azinobis-(3-ethylbenzthiazoline-6-sulfonate) (ABTS), and 6-hydroxy-2,5,7,8-tetramethylchroman-2-carboxylic acid (Trolox) with BR level were provided by Sigma Co. (St. Louis, MO, USA). The polyphenol (phloxol, gallic acid, protocatechuic acid, chlorogenic acid, 2,4-dihydroxybenzoic acid, vanillic acid, syringic acid, *p*-coumaric acid, ferulic acid, salicylic acid, benzoic acid, *o*-coumaric acid veranic acid, catechin, naringin, hesperidin, myricetin, quercetin, naringenin, kaempferol, and rutin, ≥98.0%) and anthocyanin (cyanidin, cyanidin-3-glucoside, delphinidin, delphinidin-3-glucoside, malvidin, malvidin-3-glucoside, pelargonidin, pelargonidin-3-glucoside, peonidin, peonidin-3-glucoside, petunidin, and petunidin-3-glucoside, ≥98.0%) standards were provided by Shanghai Yuanye Bio-Technology Co., Ltd. (Shanghai, China). Finally, the Folin-Ciocalteu reagent (GR) was provided by Beijing Solarbio Science & Technology Co., Ltd. (Beijing, China). Deionized water was used in the test, while chromatographic grade glacial acetic acid and acetonitrile were used for phenolic composition analysis. All of the other chemicals and reagents used in the experiments were domestic analytical pure.

### 2.2. Determination of Nutrient Composition

The nutritional composition (moisture, ash, protein, fiber, and fat content) of the different colored highland barley varieties were determined according to the methods of the Association of Official Analytical Chemists [20]. The total starch, amylose, and β-glucan content were respectively analyzed via the methods of the total starch, amylose/amylopectin, and mixed-linkage beta-glucan kits.

### 2.3. Extraction of Polyphenols

The polyphenols of highland barley with different grain colors were extracted using the previously reported method [2]. Next, 1 g of highland barley whole powder was extracted with 25 mL of 80% aqueous acetone for 20 min with the assistance of an ultrasonic clearer (KQ-500DE, Kunshan, China) at room temperature. After extraction, the supernatant was separated by centrifugating at 3000× *g* for 20 min using a TGL-20M refrigerated centrifugation (Changsha, China). The above procedure was repeated three times, and then the supernatants were merged and evaporated to dryness at 45 °C under vacuum. Finally, the residues were diluted to 10 mL with methanol and filtered with a 0.45 μm organic membrane to obtain the polyphenol extract, then stored in the dark at −20 °C until use.

### 2.4. Extraction of Anthocyanins

The highland barley anthocyanins were extracted according to a reported method with some modifications [12]. In brief, 1 g of highland barley whole powder was added to 10 mL of methanol acidified with 0.1% HCl (*v*/*v*). The mixture was left to stand for 15 min at room temperature, then further extracted in an ultrasonic cleaner (KQ-500DE, Kunshan, China) at 100 Hz for 30 min. The supernatant was collected by centrifuging at 3000× *g* for 20 min, and the residue was extracted twice following the same steps. Finally, the three extraction solutions were merged and evaporated to dryness at 45 °C under vacuum. The residues were diluted to 10 mL with deionized water and filtered with a 0.45 μm organic membrane to obtain the anthocyanin extract, then stored in the dark at −20 °C until use.

### 2.5. Determination of Total Phenol Content

The total phenol content of the extract was detected using the Folin–Ciocalteu method [21]. Next, 125 μL of extract was mixed with 500 μL deionized water and 125 μL Folin–Ciocalteu reagent in turn and reacted for 6 min at room temperature. Then, 1.25 mL of 7% Na_2_CO_3_ solution was added to the mixture, followed by 1 mL of water, for a final volume of 3 mL. The sample was kept away from light at room temperature for 1.5 h. After reaction, the absorbance of the sample was measured at 760 nm using a spectrometer (N4S, Yidian, Shanghai, China), and the total phenol content was calculated using gallic acid as the standard (mg/100 g DW).

### 2.6. Determination of Total Flavonoid Content

The total flavonoid content in the extract was detected using colorimetric method [21]. Next, 200 μL of 5% NaNO_2_ solution was added to 1 mL extract to react for 6 min, followed by the addition of 200 μL of 10% AlCl_3_·6H_2_O solution, and kept for another 6 min. Then, 2 mL of 4% NaOH solution was added to the mixture. After reacting in the dark at room temperature for 15 min, the absorbance of the sample was measured at 510 nm and the total flavonoid content was calculated using catechin as the standard (mg/100 g DW).

### 2.7. Determination of the Total Anthocyanin Content

The total anthocyanin content was measured using the method described by Ge et al. [8]. In brief, 1 mL of extract was mixed with 9 mL of pH 1 and pH 4.5 KCl buffer solution, respectively. Then, the absorbance of the sample at 510 and 700 nm under different pH conditions was measured, and the total anthocyanins content was calculated by the following formula:

Δ*A* = (A_510nm pH 1.0_ + A_700nm pH 4.5_) − (A_700nm pH 1.0_ + A_510nm pH 4.5_)
(1)

Total anthocyanin content (mg/100 g DW) = (Δ*A* × *V* × 10 × *M* × 10^5^)/(*ε* × *L* × *W*)(2)
where *V* is the total volume of the sample (mL); *M* is the molecular weight of cyaniding-3-glucoside (449.2 g/mol); *W* is the dry weight of highland barley (mg); *ε* is the molar absorptivity of cyaniding-3-glucoside (26,900 L/mol × cm); and L is the optical path length (1 cm).

### 2.8. Composition Analysis of Polyphenols

The phenolic acids and flavonoids of different colored highland barleys were analyzed by Waters 600E HPLC (WAT, Milford, MA, USA) using a Phenomenex C18 column (250 mm × 4.6 mm) and a UV–Vis detector at 280 nm. The injection volume of the sample was 20 μL, and the flow rate was 0.8 mL/min [12]. The mobile phase was distilled water with 0.1% glacial acetic acid (solvent A) and acetonitrile with 0.1% glacial acetic acid (solvent B), and their gradients were as follows: 0 min, 8% B in A; 2 min, 10% B in A; 27 min, 30% B in A; 50 min, 90% B in A; 51–56 min, 100% B in A; and 51–60 min, 8% B in A. The quantification of monomeric polyphenol was based on retention time. The content of monomeric polyphenol was calculated by the peak area, and the results were expressed in µg/g DW.

### 2.9. Composition Analysis of Anthocyanins

The anthocyanin compositions of the different types of highland barley were analyzed by means of HPLC-MS/MS (Q-Exactive, Dionex Ultimate 3000 RSLC, ThermoFisher, Waltham, MA, USA) using a Hypersil GOLD aQ column (100 mm × 2.1 mm) and mass detector. The mobile phase A was distilled water with 0.9% glacial acetic acid, and the mobile phase B was acetonitrile with 0.9% glacial acetic acid. The injection volume of the sample was 3 μL, and the flow rate was 0.3 mL/min. The elution program was as follows: 0 min, 2% B in A; 0.5 min, 2% B in A; 8 min, 50% B in A; 10 min, 90% B in A; 12 min, 90% B in A; 13 min, 98% B in A; and 15 min, 98% B in A. The mass spectrometry conditions were as follows: The positive electrospray ionization (+ESI) mode was adopted with a spray voltage of 3.5 kV. The capillary temperature and heater temperature were both 300 °C. The sheath gas (N_2_) and assist gas (N_2_) flows were 35 and 10 units/min, respectively. The scanning mode was full scan with a resolution of 70,000 and scanning range of 100–1500 *m*/*z*. The anthocyanin extract was filtered using 0.22 μm acrodisc syringe filter before being detected by HPLC-MS/MS. Anthocyanin standards were used for qualitative/quantitative analysis, and the results were expressed in µg/g DW.

### 2.10. DPPH· Radical-Scavenging Capacity Assay

DPPH· radical-scavenging capacity was assayed according to the method described by Bakar et al., with some modifications [22]. In a test tube, 1 mL sample and 4.5 mL 0.1 mmol/L DPPH–methanol solution were thoroughly mixed, then kept in the dark for 30 min. The absorbance of the sample was recorded at 517 nm using methyl alcohol for the blank zero setting instead of extract. The DPPH·radical-scavenging capacity of the extract was calculated according to the standard curve (Y = 0.0042X + 0.9163 (0–140 μmol/L, R^2^ = 0.9928)), and the results were expressed in μmol Trolox eq./100 g DW.

### 2.11. Ferric Reducing Antioxidant Power Assay

Ferric reducing antioxidant power (FRAP) was determined using the method described by Benzie et al., with some modifications [23]. The FRAP working solution was composed of 300 mmol/L pH 3.6 sodium acetate buffer solution, 10 mmol/L TPTZ solution, and 20 mmol/L FeCl_3_ solution (10:1:1, *v*/*v*/*v*). The working solution was ready-made and preheated in a water bath at 37 °C prior to use. In most cases, 50 µL sample and 4.5 mL FRAP working solution were mixed in a test tube and reacted in the dark for 30 min after thorough shaking. Then, the absorbance was recorded at 593 nm using methyl alcohol as a blank. The FRAP was calculated according to the standard curve (Y = 0.0072X − 0.0012 (0–300 μmol/L, R^2^ = 0.9992)), and the results were expressed in μmol Trolox eq./100 g DW.

### 2.12. ABTS·^+^ Radical-Scavenging Capacity Assay

ABTS·^+^ radical-scavenging capacity was determined according to the report by Guo et al. [24]. The ABTS·^+^ working solution was prepared by mixing 5 mL 7 mmol/L ABTS solution with 88 µL 140 mmol/L potassium persulfate solution and was then kept in the dark for 12–16 h. The stock solution was diluted to a UV–Vis absorbance of 0.7 ± 0.02 using anhydrous methanol (1:100, *v*/*v*) before use. Typically, 200 µL sample and 4.0 mL ABTS·^+^ working solution was thoroughly mixed and reacted in the dark for 30 min. Then, the absorbance was recorded at 734 nm using methyl alcohol as a blank. Finally, the ABTS·^+^ radical-scavenging capacity of the extract was calculated according to the standard curve (Y = −0.001X + 0.6242 (0–300 μmol/L, R^2^ = 0.9907)), and the results were expressed in μmol Trolox eq./100 g DW.

### 2.13. Statistical Analysis

The data were measured in triplicate and reported as mean ± SD (standard deviation). The mean, range, and variable coefficient (CV) of the data were analyzed using Excel 2003 (Microsoft, Redmond, WA, USA), and related images were drawn using Origin 2019 (OriginLab, Northampton, MA, USA). Variance and significance differences among the means were calculated using a SNK-q test, and the correlation coefficients were obtained from a Pearson test. Finally, classification analysis was conducted using SPSS 22.0 (SPSS Inc., Chicago, IL, USA). Statistical significance and extreme significance were defined as *p* < 0.05 and *p* < 0.01, respectively.

## 3. Results and Discussion

### 3.1. Analysis of Nutritional Composition

It can be seen from Table 2 that the difference in the average content of nutritional components of highland barley in the purple, blue, black, and yellow groups was generally small. The average protein content of black and purple highland barley was the highest, without significant difference (*p* > 0.05). More specifically, the varieties of black highland barley with protein content > 13% accounted for 75% varieties, followed by the varieties of purple highland barley, for which varieties with protein content > 12% accounted for 75%. Yellow and blue highland barley had the lowest protein contents, without significant difference (*p* > 0.05), and their varieties with protein content >11% accounted for 50% of varieties. The varieties with the highest protein content in purple, blue, black, and yellow groups were No. 2, No. 19, No. 32, and No. 51, respectively. The protein content of highland barley with different grain colors ranged from 8.14% to 14.70%, with an average of 12.10%, which was consistent with the results reported by Wirkijowska et al. This is generally higher than those of barley, wheat, corn, oats, and other grains [25]. Zheng et al. measured the nutritional composition of six highland barley varieties and found that the crude protein content ranged from 9.2% to 10.2%. More specifically, the respective protein contents of Zangqing 25, Beiqing 6, and Dulihuang highland barley were 9.4%, 10.1%, and 10.1%, respectively. The exception was Zangqing 25, which had a value that was lower than the test results of corresponding varieties in this study; this may have been a result of different planting locations [26].

The fat content of highland barley with different grain colors ranged from 1.64% to 2.40%, with an average of 1.85%. The varieties with fat content higher than 2% accounted for 23% of the total varieties tested. Blue highland barley had the highest crude fat content, with an average content of 2.00%, while black highland barley had the lowest crude fat content of 1.80%; yellow, black, and purple varieties had average fat contents of about 1.8%, without significant difference (*p* > 0.05). The varieties with the lowest fat content in the purple, blue, black, and yellow groups were No. 4, No. 19, No. 35, and No. 46, respectively. It was reported that the fat content of highland barley was in the range of 2.01–3.09% (average 2.13%), which was lower than those of corn, sorghum and oats, but higher than that of rice [19,25]. In this study, the average fat content of highland barley with different grain colors was significantly lower than the reported results. It can be seen that highland barley grown on the Qinghai Tibet Plateau, particularly the four varieties with the lowest fat content, generally belongs to the group of high-quality, low-fat food processing raw materials.

The total starch content in the blue group > yellow group > black group > purple group, without significant difference (*p* > 0.05). The average total starch content of highland barley with different grain colors was 32.57–70.00%, with an average value of 58.37%, which was lower than those of wheat, corn, rice and barley [9,27]. The highland barley varieties with the highest total starch content in the purple, blue, black, and yellow groups were No. 11, No. 18, No. 29, and No. 52, respectively. The amylose content of highland barley with different grain colors ranged from 14.80% to 28.28%, with an average content of 21.87%. There was no significant difference between the purple and black groups (*p* > 0.05), nor between the yellow and blue groups (*p* > 0.05). However, the amylose contents of the purple group and the black group were significantly higher than those of the yellow and blue groups (*p* < 0.05). The highland barley varieties with the highest amylose contents in the purple, blue, black, and yellow groups were No. 11, No. 23, No. 34, and No. 50, respectively. Mangan et al. analyzed the nutritional characteristics of 33 highland barley resources, and their results showed that the amylose content ranged from 11.70% to 25.0%, which was basically consistent with the results of the present study [28]. The chemical and functional properties of highland barley starch depend on the content, proportion, and particle structure of amylose and amylopectin [29], and the starch content and composition are the most important factors affecting the processing technology and product quality of highland barley. High total starch content may be suitable for fermentation processing, while low amylose content is suitable for processing into some foods with soft texture [10].

The crude fiber content of highland barley with different grain colors was 1.81–3.64%, with an average of 2.62%, which was close to the fiber content of Tibetan highland barley as reported by Zhang et al. (2.76–3.02%) [27]. The average content of crude fiber in the yellow highland barley was the highest (2.76%), while that in blue highland barley was the lowest (2.37%). The average content of crude fiber in the purple, blue, and black groups was 2.37–2.57% without significant difference (*p* > 0.05), thus indicating that ordinary yellow highland barley grown on the Qinghai Tibet plateau may contain higher concentrations of dietary fiber. The varieties with the highest crude fiber contents in the purple, blue, black, and yellow groups were No. 13, No. 19, No. 34, and No. 52, respectively. These varieties are good raw materials for processing into highland barley food, as they are rich in dietary fiber.

The ash content of highland barley with different grain colors ranged from 0.04% to 0.94% with an average content of 0.41%, which was lower than those of naked barley, wheat, and oats [25,30,31]. The average ash content of highland barley in the black group was the highest (0.64%), which was significantly higher than that in the blue group (*p* < 0.05). The average ash content of highland barley in the purple and yellow groups was relatively low, without significant difference (*p* > 0.05). The varieties of highland barley with the highest ash content in the purple, blue, black, and yellow groups were No. 3, No. 23, No. 34, and No. 49, respectively. The overall variation coefficient of ash content of highland barley with different grain colors reached 67.18%. That is to say there were great differences in ash content among 52 highland barley varieties, thus indicating that variety bore an important impact on the ash content of highland barley. Ash content can predict the level of minerals in food, thus indicating that the mineral content of highland barley with different grain colors selected in this study may have been low.

β-glucan content of highland barley with different grain colors was between 3.88–6.88%, with an average of 5.22%, which was consistent with the content of 3.66–8.62% reported in previous studies. The average β-glucan content of black highland barley was the highest (6.88%); the average β-glucan content of yellow highland barley was the lowest (3.88%); and there was no significant difference in the average β-glucan content among the black, blue, and purple highland barleys (*p* > 0.05), which were much higher than that of common wheat [10]. Among all the highland barley tested, the varieties with β-glucan content > 5% accounted for 51.92% of varieties, indicating that highland barley is rich in β-glucan—especially the dark (black, purple, and blue) highland barleys, which have even higher β-glucan contents [13]. The highland barley varieties with the highest β-glucan content in the purple, blue, black, and yellow groups were No. 11, No. 15, No. 38, and No. 41, respectively. Highland barley β-glucan is an active polysaccharide located in the cell wall and aleurone layer of grain endosperm which contains glucopyranose; it plays an important role in controlling blood glucose, reducing cholesterol level, and enhancing immunity [32,33]. It can be seen that highland barley is an ideal source of raw materials for processing into healthy food with high β-glucan content.

**Table 2 foods-11-02025-t002:** Analysis on nutritional quality of different colored highland barley varieties (g/100 g DW).

Color	Number		Protein	Fat	Fiber	Total Starch	Amylose	Ash	β-glucan
Purple	14	Mean	12.69 ± 1.81 ^a^	1.85 ± 0.15 ^a,b^	2.55 ± 0.29 ^a,b^	55.21 ± 8.29 ^a^	24.72 ± 2.82 ^a^	0.28 ± 0.18 ^c^	5.26 ± 1.12 ^a^
		Range	8.14 (No.13)–14.70 (No.2)	1.65 (No.4)–2.16 (No.13)	2.06 (No.11)–3.02 (No.13)	32.57 (No.10)–61.69 (No.11)	17.99 (No.12)–28.28 (No.11)	0.04 (No.6)–0.69 (No.3)	4.47 (No.2)–6.08 (No.11)
		CV/%	14.26	8.11	11.39	15.02	11.41	64.30	8.27
Blue	12	Mean	11.31 ±1.84 ^b^	2.00 ± 0.21 ^a^	2.37 ± 0.40 ^b^	60.68 ± 3.96 ^a^	19.92 ± 2.79 ^b^	0.40 ± 0.22 ^b^	5.29 ± 1.43 ^a^
		Range	8.51 (No.23)–13.85 (No.19)	1.75 (No.19)–2.40 (No.16)	1.81 (No.23)–3.02 (No.19)	54.10 (No.19)–68.62 (No.18)	16.26 (No.16)–24.04 (No.23)	0.07 (No.22)–0.84 (No.23)	4.21 (No.26)–6.06 (No.15)
		CV/%	16.27	10.50	16.88	6.52	14.01	55.00	14.31
Black	12	Mean	13.20 ± 0.70 ^a^	1.80 ± 0.18 ^b^	2.57 ± 0.45 ^a,b^	57.89 ± 3.70 ^a^	22.65 ± 2.05 ^a^	0.64 ± 0.24 ^a^	5.29 ± 0.98 ^a^
		Range	11.74 (No.30)–14.07 (No.32)	1.59 (No.35)–2.11 (No.33)	1.99 (No.38)–3.64 (No.34)	52.81 (No.36)–63.37 (No.29)	18.96 (No.32)–25.94 (No.34)	0.16 (No.31)–0.94 (No.34)	3.96 (No.28)–6.88 (No.38)
		CV/%	5.30	10.00	17.50	6.40	9.05	37.50	23.68
Yellow	14	Mean	11.21 ± 1.83 ^b^	1.82 ± 0.13 ^a,b^	2.76 ± 0.37 ^a^	58.4 ± 5.73 ^a^	20.34 ± 2.52 ^b^	0.25 ± 0.17 ^c^	5.04 ± 1.03 ^b^
		Range	8.49 (No.49)–12.76 (No.51)	1.64 (No.46)–2.06 (No.42)	2.12 (No.41)–3.28 (No.52)	49.14 (No.41)–68.22 (No.52)	14.80 (No.52)–24.05 (No.50)	0.04 (No.40)–0.54 (No.49)	3.88 (No.40)–6.78 (No.41)
		CV/%	16.30	7.14	13.40	8.80	12.39	68.00	18.84
Total	52	Mean	12.10 ± 1.80	1.85 ± 0.19	2.62 ± 0.36	58.37 ± 7.16	21.87 ± 3.14	0.41 ± 0.27	5.22 ± 0.81
		Range	8.14–14.70	1.64–2.40	1.81–3.64	32.57–68.62	14.8–28.28	0.04–0.94	3.88–6.88
		CV/%	14.86	10.23	13.79	8.04	14.35	67.18	15.56

Note: Different letters after data within the same column indicate significant differences (*p* < 0.05). The numbers in parentheses are highland barley variety numbers. CV: coefficient of variation.

### 3.2. Analysis of Phenolic Content

It can be seen from Table 3 that the average content of phenolic acids in highland barley with different grain colors was distributed in the range of 204.29–225.16 mg/100 g DW; black group > yellow group > blue group > purple group. However, there was no significant difference between the black group and the yellow group (*p* > 0.05), nor between the purple and yellow groups (*p* > 0.05). There was no significant difference in the average content of flavonoids in the blue, black, and yellow groups (23.33–23.78 mg/100 g DW) (*p* > 0.05), while the content of flavonoids in the purple group was the lowest (19.88 mg/100 g DW). The average content of total phenols was 8.80–10.28 times that of total flavonoids, thus indicating that there were few flavonoids in highland barley polyphenols. Yang et al. studied the content and composition of phenolic compounds in blue highland barley and found that the polyphenols in blue highland barley were mainly phenolic acids in bound and free states [2]. Ge et al. observed that among white, yellow, black, and blue highland barley, black and white highland barley had higher contents of total phenolic acids and total flavonoids, which was similar to the results of the present study [8]. The varieties with the highest contents of total phenols in the purple, blue, black, and yellow groups were No. 12, No. 23, No. 28, and No. 50, respectively, while the varieties with the highest content of total flavonoids were No. 13. No. 26, No. 29, and No. 50, respectively. Highland barley in the different grain color groups contained varieties with significantly higher or lower contents of phenolic compounds, thus indicating that the difference between highland barley varieties was also an important factor affecting the content of polyphenols [8]. The content of total anthocyanins in highland barley with different grain colors was 12.94–18.65 mg/100 g DW. The content of total anthocyanins in the purple group was significantly higher than that in the other groups (*p* < 0.05), while those in the blue and black groups were the same and significantly higher than that in the yellow group, respectively (*p* < 0.05). The varieties with the highest anthocyanin contents in the purple, blue, black, and yellow groups were No. 13. No. 15, No. 36, and No. 39, respectively. Colored (purple, black, blue, red, etc.) grains are rich in anthocyanins, and are considered to be promising ingredients for the development of whole grain functional foods [1]. Previous research showed that the anthocyanin content of barley grains was directly proportional to the color depth. Blue and purple barley grains had been proven to have the highest anthocyanin content among all barley varieties, which was consistent with the detection results of the present study [34]. In summary, the polyphenols of highland barley are mainly phenolic acids, while flavonoids and anthocyanins appear in lower quantities. Moreover, grain color and variety of highland barley exert different effects on its phenolic compounds. In the large sample group, genotype bears a relatively large impact on the overall level of polyphenols in highland barley, while grain color bears a relatively large impact on the overall level of anthocyanins. Phenolics have an impact on the taste, color and health of food. When we choose highland barley as the raw material for functional food processing, we should comprehensively consider the interaction between its grain color and highland barley genotype [11,12,31].

### 3.3. Composition Analysis of Phenolic Acids and Flavonoids

According to Table 4, the average content of total phenolic acids of highland barley with different grain colors was measured by the HPLC method was 115.44–244.08 µg/g, and the order of content was black > purple > yellow > blue, without significant differences among the different grain color groups (*p* < 0.05). The average content of total flavonoids was 65.33–90.10 µg/g, and the order of content was yellow > black > purple > blue, without significant differences between the purple and black groups (*p* > 0.05). The content of total phenolic acids was significantly higher than that of total flavonoids (*p* < 0.05), and it is further confirmed that soluble phenolic acids were the main form that colored highland barley polyphenols appeared in [2,19]. The average content of total phenols in highland barley with different grain colors were significantly different (*p* < 0.05), and the order from high to low was black (329.93 µg/g) > purple (271.36 µg/g) > yellow (245.07 µg/g) > blue (180.77 µg/g), indicating that black barley contained more abundant phenolic compounds. This result was consistent with that of the chemical method (refer to Table 3). Ge et al. detected 89 phenolic compounds in black highland barley, which was the most abundant of the four groups of highland barley with different grain colors (white, yellow, blue, and black), and this result was consistent with the findings of the present study [8]. However, the content of total phenols in highland barley determined by the chemical method and HPLC method was different among the purple, blue, and yellow groups. A possible reason for this is that the polyphenols determined by chemical method are mixed components, and the results are presented in terms of relative content of gallic acid or catechin.

In terms of monomeric polyphenol, benzoic acid (42.73 µg/g) was the highest average phenolic acid in the purple group, phloxol (82.05 µg/g) was the richest phenolic acid in the black group, and chlorogenic acid was the most abundant phenolic acid in the blue and yellow groups. The average contents of protocatechuic acid (22.93–40.88 µg/g), chlorogenic acid (28.32–50.56 µg/g), and gallic acid (9.11–20.47 µg/g) in highland barley with different grain colors were high, and their total accounts for 32.51–65.52% of the total phenolic acids—followed by vanillic acid, veratric acid, and *p*-coumaric acid—indicating that these phenolic acids were characteristic phenolic acids commonly found in highland barley [35,36]. The contents of 2,4-dihydroxybenzoic acid, syringic acid, and ferulic acid in the four groups of highland barley with different colors were lower. Catechin (33.09–46.94 µg/g) had the highest content in highland barley of the different grain color groups, which was significantly higher than other flavonoids, accounting for 45.11–52.10% of the total flavonoids—followed by kaempferol, rutin, and naringenin—thus indicating that these compounds were the main characteristic flavonoids in highland barley [2,35,37]. The contents of naringin, hesperidin, and myricetin in highland barley were low (0.47–6.63 µg/g); hesperidin was not detected in the yellow group, and myricetin and quercetin were not detected in the blue group. The average content of quercetin was higher only in the yellow group (12.21 µg/g). The results of the study by Shen et al. showed that ferulic acid was the most abundant polyphenol in highland barley of the Zangqing2000 (black), Xunhua (blue), and Shangri-La (green) varieties, followed by *p*-coumaric acid [37]. Abdel-Aal et al. reported that the most abundant free and bound polyphenols in black, blue, and yellow barley were ferulic acid, followed by *p*-coumaric acid [11]. Kim et al. found that the main flavonoid of black, blue, and purple barley was myricetin, while the contents of naringin and hesperidin were quite low [12]. The above reports are inconsistent with the results of the present study. This inconsistency may be related to the different genotypes, planting environments, and extraction methods of barley or highland barley varieties used in the experiment and reveals that genotype and grain color bear an important impact on the composition and content of polyphenols in highland barley [11].

### 3.4. Composition Analysis of Anthocyanins

It can be seen from Table 5 that the composition of anthocyanins in highland barley with different grain colors was similar, yet the content was significantly different (*p* < 0.05). Nine types of anthocyanins were detected in highland barley in the purple, blue, black, and yellow groups, and the order of average content was purple > blue > black > yellow, which was basically consistent with the results determined by the chemical method. The average contents of anthocyanins in purple, blue, and black highland barley were respectively 10.49, 4.50, and 3.77 times higher than that in yellow highland barley, thus indicating that dark highland barley was more abundant in anthocyanins than ordinary yellow highland barley [38]. Cyanidin-3-glucoside (30.83 μg/g) had the highest anthocyanin content of highland barley in the purple group, followed by delphinidin-3-glucoside, accounting for 68.63% and 20.17% of the total anthocyanins, respectively. Delphinidin-3-glucoside (10.94 and 7.37 μg/g) had the highest anthocyanin content of highland barley in the blue and black groups, followed by cyanidin-3-glucoside. Delphinidin-3-glucoside and cyanidin-3-glucoside accounted for 56.74% and 13.54%, respectively, of the total in the blue group and 45.66% and 30.42%, respectively, in the black group. Delphinidin (1.06 μg/g) had the highest anthocyanin content of highland barley in the yellow group, followed by cyanidin-3-glucoside, accounting for 24.77% and 21.26% of the total, respectively. Kim et al. analyzed the anthocyanin contents and compositions of purple, blue, and black barley resources. The results showed that the anthocyanin contents of purple and blue barley were significantly higher than that of black barley. Cyanidin-3-glucoside was the most abundant anthocyanin in purple barley, while delphinidin-3-glucoside was the most abundant anthocyanin in blue and black barley, which was similar to the findings of the present study [12]. Anthocyanins are a class of flavonoid compounds with good antioxidant activity and have biological activities such as antibacterial, anti-inflammatory, antihypertensive, anticancer and antiobesity activities [39]. Therefore, it is indicated by the results of this study that dark highland barley, particularly purple highland barley, has great potential in the development of functional food.

### 3.5. Antioxidant Capacity of Polyphenols and Anthocyanins

As shown in Figure 1, the DPPH·-scavenging capacity, FRAP reducing power, and ABTS·^+^-scavenging capacity of highland barley polyphenols and the DPPH·-scavenging capacity, FRAP-reducing power, and ABTS·^+^-scavenging capacity of highland barley anthocyanins were all similar among the various groups. This showed that it was difficult to distinguish the antioxidant capacity of highland barley groups by grain color alone. Highland barley anthocyanins had stronger DPPH·-scavenging capacity, while highland barley polyphenols had stronger FRAP ferric-ion-reducing capacity and ABTS·^+^-scavenging capacity. That is to say polyphenols and anthocyanins had different contributions to the antioxidant capacity of the three antioxidant systems. The overall antioxidant capacity of black highland barley polyphenols was the strongest, yet there was no significant difference between black highland barley and yellow highland barley (*p* > 0.05). The overall antioxidant capacity of purple highland barley was lower than only black highland barley and yellow highland barley, while the overall antioxidant capacity of blue highland barley was the weakest. The main reason for this phenomenon was that polyphenols content and composition in highland barley with different grain colors were different. Among them, black highland barley was generally richer in polyphenols compounds, while blue highland barley was usually the least rich (refer to Table 3 and Table 4). Shen et al. found that the DPPH·-free-radical-scavenging capacity, ferric reducing power and ORAC values of free phenols in black highland barley (Zangqing 2000) were significantly higher than those in blue highland barley (Xunhua) and green highland barley (Shangri-La) (*p* < 0.05) [37]. The research results of Tang et al. showed that the DPPH·-scavenging capacity and FRAP-reducing power of black highland barley (Changheiqingke) were higher than those of blue highland barley (Dulihuang) [40]. These results are consistent with the test results of this study.

The overall antioxidant capacity of purple highland barley anthocyanins was the strongest, followed by black highland barley, while the respective overall antioxidant capacities of blue and yellow highland barley were the same. The content and composition of anthocyanins in highland barleys with different grain colors exerted an effect on their antioxidant capacities. Table 3 and Table 5 show that purple highland barley was the most abundant in anthocyanins, followed by blue and black highland barley (in terms of average content); the anthocyanin content of yellow highland barley was the lowest, which was consistent with the order of antioxidant capacity of the four groups of highland barley. Lee et al. previously reported the antioxidant capacity of anthocyanin extracts from four different genotypes of colored barley. The results showed that the order of total antioxidant capacity was purple barley (Yu 5904-088) > blue barley (Ubamer) > black barley (black) > yellow highland barley (Meresse) [39]. This result was different from those of the present study in that the overall antioxidant capacity of black highland barley anthocyanins was stronger than that of blue highland barley. The main reason for this lies in the fact that Lee et al. applied a single genotype of barley for comparative study, while in the present study, group highland barley samples were used for comparative study. Therefore, the impact of genotype and variety differences could not be ignored. In conclusion, the composition and content of phenolic substances in highland barley with different grain colors exert a great impact on the antioxidant capacity. Different phenolic compounds bear certain selectivity for the scavenging of different free radicals, which eventually leads to the difference of antioxidant capacity between single varieties or colored groups [11,41].

### 3.6. Correlation between Nutritional Quality and Functional Quality

According to Table 6, the highest and extremely significant positive correlation was between ash content and rutin content (r = 0.4480, *p* < 0.01). There was a significant (*p* < 0.05) or extremely significant (*p* < 0.01) correlation between crude fiber content and TPC, TPH, veratric acid, naringin, and hesperidin content. There was a significant positive correlation between crude fat content and *p*-coumaric acid content (*p* < 0.05). The crude protein content was positively correlated with the content of TPAH, TPH, gallic acid, kaempferol, and syringic acid (*p* < 0.05) and was negatively correlated with the content of cyanidin, delphinidin, and peonidin (*p* < 0.05). The amylose content was also significantly related to the levels of TFC, TPAH, TPH, TANH, veratric acid, kaempferol, naringenin, naringin, myricetin, cyanidin-3-glucoside, delphinidin, malvidin-3-glucoside, and pelargonidin-3-glucoside. In addition, there was a significant positive correlation between the content of β-glucan and the content of *p*-coumaric acid, kaempferol, delphinidin-3-glucoside, and peonidin (*p* < 0.05). The results indicated that there was a close relationship between the nutritional quality and functional quality of highland barley and that the latter can be reflected by the former. Interestingly, the crude fiber content can be used to achieve the preliminary prediction of total phenol content, and the crude protein content can be used to complete the preliminary prediction of phenolic acid content; the resistant starch content can be used to perform preliminary prediction of total anthocyanin content.

Tong et al. reported that there was a significant negative correlation between the content of ferulic acid and the content of crude protein, crude fat, and ash in oats (*p* < 0.05), while there was an extremely significant positive correlation between the content of total phenols and the contents of crude protein and ash (*p* < 0.01) [42]. Suriano et al. found that there was a significant positive correlation between the content of β-glucan and the contents of total phenols and crude protein in Italian colored barley (*p* < 0.05) [1]. Memon et al. reported that there were correlations of varying degrees between the contents of crude fiber, crude fat, crude protein, ash, *p*-coumaric acid, syringic acid, and gallic acid in three different wheat varieties (Benazir, TJ-83 and Imdad), and there were also obvious differences between these varieties. The content of crude fiber, crude fat, and ash in the wheat three varieties were positively correlated with the contents of *p*-coumaric acid, syringic acid, and gallic acid overall, while the crude protein content was positively correlated with the contents of these three phenolic acids only in Benazir wheat. There was a negative correlation between crude protein content and crude fat content in TJ-83 and Imdad wheat, but a positive correlation between crude protein content and crude fat content in Benazir wheat [43]. These research reports differ from the results of this study, indicating that the relationship between nutritional and functional quality also differs due to the different types and varieties of crops. In summary, the content of various functional components in different highland barley varieties can be evaluated from the level of main nutrients, and this conclusion bears guiding significance for the selection and breeding of improved highland barley varieties.

### 3.7. Correlation between Polyphenol Content and Antioxidant Capacity

As shown in Table 7, there was either a significant (*p* < 0.05) or extremely significant positive correlation (*p* < 0.01) between TPC, TFC, and TPH and DPPH·-scavenging capacity, ABTS·^+^-scavenging capacity, and FRAP-reducing power. TPAH had a significant positive correlation with DPPH·-scavenging capacity and ABTS·^+^-scavenging capacity (*p* < 0.05). The contents of protocatechuic acid and catechin showed a significant positive correlation with DPPH·-scavenging capacity. The contents of chlorogenic acid and catechin showed a significant positive correlation with FRAP reducing power (*p* < 0.05). Benzic acid content showed an extremely significant correlation with ABTS·^+^-scavenging capacity (*p* < 0.01). These results indicated that the content of polyphenols in highland barley with different grain colors significantly related to their antioxidant capacity. Protocatechuic acid and catechin were the main contributors to the antioxidant capacity of DPPH. Chlorogenic acid and catechin were the main contributors to the antioxidant capacity of FRAP. Benzoic acid was the main contributor to the antioxidant capacity of ABTS.

Boubakri et al. analyzed the phenolic components and antioxidant capacity of Tunisian barley. Their results showed that there was a strong correlation between the content of total phenols and DPPH·-scavenging capacity, ABTS·^+^-scavenging capacity and FRAP-reducing power, while catechin-3-glucose and hydroferuloyl were the main contributors to the antioxidant effect of various antioxidant systems [17]. Yang et al. reported that hydroxybenzoic acid and protocatechuic acid in blue highland barley were the main contributors to DPPH·- and ABTS·^+^-scavenging capacity [2]. Abdel-Aal et al. held that *p*-coumaric acid in barley polyphenols was the main contributor to the scavenging of DPPH·, and that ferulic acid and vanillic acid were the main contributors to the scavenging of ABTS·^+^ [11]. In the present study, the reason for the differences in the main contribution of monomeric polyphenols in the various antioxidant systems may have been that the composition and content of polyphenols differed due to the different species and growth environment, in turn resulting in the various types of antioxidant substances [11,21].

### 3.8. Correlation between Anthocyanin Content and Antioxidant Capacity

It can be seen from Table 8 that TANC and TANH exhibited either a significant (*p* < 0.05) or extremely significant (*p* < 0.01) positive correlation with DPPH·-scavenging capacity, FRAP-reducing power and ABTS·^+^-scavenging capacity. The content of cyanidin-3-glucoside, malvidin-3-glucoside, and pelargonidin-3-glucoside had a significant positive correlation with DPPH·-scavenging capacity (*p* < 0.05). The contents of cyanidin, cyanidin-3-glucoside, delphinidin, and pelargonidin-3-glucoside had either a significant (*p* < 0.05) or extremely significant (*p* < 0.01) positive correlation with FRAP-reducing power. The contents of cyanidin-3-glucoside and delphinidin-3-glucoside had a significant positive correlation with ABTS·^+^-scavenging capacity (*p* < 0.05). The results indicated that the content of anthocyanins in highland barley with different grain colors also significantly related to their antioxidant capacity. Cyanidin-3-glucoside, malvidin-3-glucoside, and pelargonidin-3-glucoside were the main contributors to the antioxidant capacity of DPPH. Cyanidin, cyanidin-3-glucoside, delphinidin, and pelargonidin-3-glucoside were the main contributors to the antioxidant capacity of FRAP. Cyanidin-3-glucoside and delphinidin-3-glucoside were the main contributors to the antioxidant capacity of ABTS.

Kim et al. reported that the DPPH·-scavenging capacity of seven colored barley varieties was highly positively correlated with the content of polyphenols and proanthocyanidins, and the correlation coefficients between DPPH·-scavenging capacities and contents of proanthocyanidins, total phenols, total phenolic acids, and total flavonoids were 0.56, 0.37, 0.38, and 0.21, respectively [12]. The research results of Suriano et al. showed that there was a significant positive correlation between the content of total proanthocyanidins in 20 colored barley varieties in southern Italy and the scavenging capacities of DPPH· and ABTS·^+^, and the Pearson correlation coefficients were 0.30 and 0.31, respectively [1]. Lin et al. reported that there was an extremely significant positive correlation between the contents of total proanthocyanidins and total anthocyanins of Tibetan highland barley and DPPH·-scavenging capacity, FRAP-reducing power and ABTS·^+^-scavenging capacity (*p* < 0.01), and the Pearson correlation coefficients were in the range of 0.8878–0.9906 [13]. The above reports are basically consistent with the findings of the present study.

### 3.9. Classification Analysis of Highland Barley Varieties

In order to more clearly observe the quality characteristics of highland barley with different grain colors, the intergroup connection method was used to classify and analyze nutritional quality and functional quality of the highland barley varieties tested according to the Euclidean distance. The results were shown in Figure 2. Figure 2A shows that the 52 highland barley varieties could be divided into four categories according to various levels of nutritional quality, and there were certain similarities among the same cluster varieties. The first category included 18 highland barley varieties, including five types of purple highland barley, four types of blue highland barley, six types of black highland barley, and three types of yellow highland barley. The second category included 11 varieties, including four types of blue highland barley, one type of black highland barley, and six types of yellow highland barley. The third category included nine highland barley varieties, including four types of purple highland barley, two types of blue highland barley, two types of black highland barley, and one type of yellow highland barley. Finally, the fourth category included 14 highland barley varieties, including five types of purple highland barley, two types of blue highland barley, three types of black highland barley, and four types of yellow highland barley. Among all of the categories, the average content of crude protein in the first category of highland barley was the highest (13.02%), and the average content of β-glucan (5.14%) and total starch (58.95%) was at the middle level, which was suitable for processing highland barley food rich in nutrients. The average content of total starch in the second category of highland barley was high (61.61%), but the average contents of β-glucan (4.64%), amylose (18.91%), and ash (0.33%) were the lowest in all clusters, which is suitable for processing into some highland barley foods with good organoleptic quality. The average contents of total starch (64.87%), amylose (24.69%), and ash (0.49%) in the third category of highland barley were the highest, and it was shown to be rich in β-glucan (5.40%), thus making it suitable for processing fermented highland barley products. The average content of β-glucan (5.66%) in the fourth category of highland barley was the highest, and the average content of total starch was the lowest (52.75%); it was rich in crude protein (12.80%), thus making it suitable for processing highland barley food rich in β-glucan.

Figure 2B shows that the 52 highland barley varieties could be divided into four categories according to different functional characteristics. The first category included 18 highland barley varieties, including five types of purple highland barley, three types of blue highland barley, six types of black highland barley (accounting for the largest proportion), and four types of yellow highland barley varieties. The second category included 21 highland barley varieties, including nine types of purple highland barley (accounting for the largest proportion), three types of blue highland barley, six types of black highland barley, and three types of yellow highland barley. The third category included seven highland barley varieties, including two types of blue highland barley and five types of yellow highland barley (accounting for the largest proportion). Finally, the fourth category included six highland barley varieties, including four types of blue highland barley (accounting for the largest proportion) and two types of yellow highland barley. According to the functional quality, the discrimination of four groups of highland barley with different grain color was more obvious, thus indicating that the grain color of highland barley had a certain impact on its functional quality, which was consistent with the results of previous analysis. Among all categories, the first category of highland barley had a high average content of polyphenols (230.08 μg/g) and anthocyanins (24.86 μg/g) and strong antioxidant capacity, with the best functional quality. It was suitable for processing highland barley food rich in phenolics and with strong antioxidant capacity. The second category of highland barley had the highest average content of polyphenols (232.52 μg/g) and anthocyanins (25.43 μg/g). There was no significant difference between the second category and the first category (*p* > 0.05), but the antioxidant capacity of the former was lower than that of the first category; its functional quality was second only to the first category. The third category of highland barley had the lowest average content of polyphenols (211.65 μg/g) and anthocyanins (7.69 μg/g), and its antioxidant capacity was also significantly lower than that of the first and second categories. It belonged to a group of highland barley varieties with relatively common functional quality. The average content of polyphenols (227.44 μg/g) in the fourth category of highland barley was high, while the average content of anthocyanins (13.05 μg/g) was low. The total antioxidant capacity was significantly lower than that of other categories of highland barley. It was suitable for processing highland barley food rich in polyphenols.

According to the results of Figure 2A,B, 13Y11-5 and Ganziheiliuleng had relatively higher polyphenol contents, total anthocyanin contents, and antioxidant capacities, and their functional qualities were outstanding. Meanwhile, Z560, Kunlun 17, and Kunlun 20 had higher total starch content, crude protein content, and β-glucan content, and their nutritional qualities were outstanding. Ganziheiliuling had higher polyphenol content, total anthocyanin content, antioxidant capacity, crude protein content, and β-glucan content, and its comprehensive quality was quite high. In general, each category includes highland barley varieties with different grain colors, thus indicating that grain color cannot be used as an absolute index to screen and evaluate the value of highland barley. Although dark highland barley is richer in anthocyanin content, it does not necessarily have high nutritional value. The variety difference caused by genotype is the main factor affecting the excellent quality of highland barley. At the same time, appropriate raw materials must be selected according to the processing purpose and variety of characteristics of highland barley in actual production.

## 4. Conclusions

The difference between nutritional compositions of highland barley in different grain color groups is small. Overall, they have lower contents of crude fat and ash and higher contents of crude protein and carbohydrates. All dark highland barley is richer in β-glucan. The highland barley with different grain colors has high antioxidant capacity. Among the four highland barley groups, the polyphenol content and antioxidant capacity of black highland barley are higher, and the anthocyanin content and antioxidant capacity of purple highland barley are the highest. In addition, the determination of main nutrients can be used to roughly evaluate the content and composition of highland barley phenolics, which provides a simple method for the selection of excellent highland barley varieties. Highland barley can be clearly classified according to nutritional quality and functional quality. Dark variety highland barley in a large sample group does not necessarily possess superior nutritional and functional quality. Variety is the main factor affecting the quality characteristics of highland barley. Suitable varieties should be selected according to their use in production. These results can provide a theoretical basis for the breeding of improved highland barley varieties, healthy consumption, and food development.

## Figures and Tables

**Figure 1 foods-11-02025-f001:**
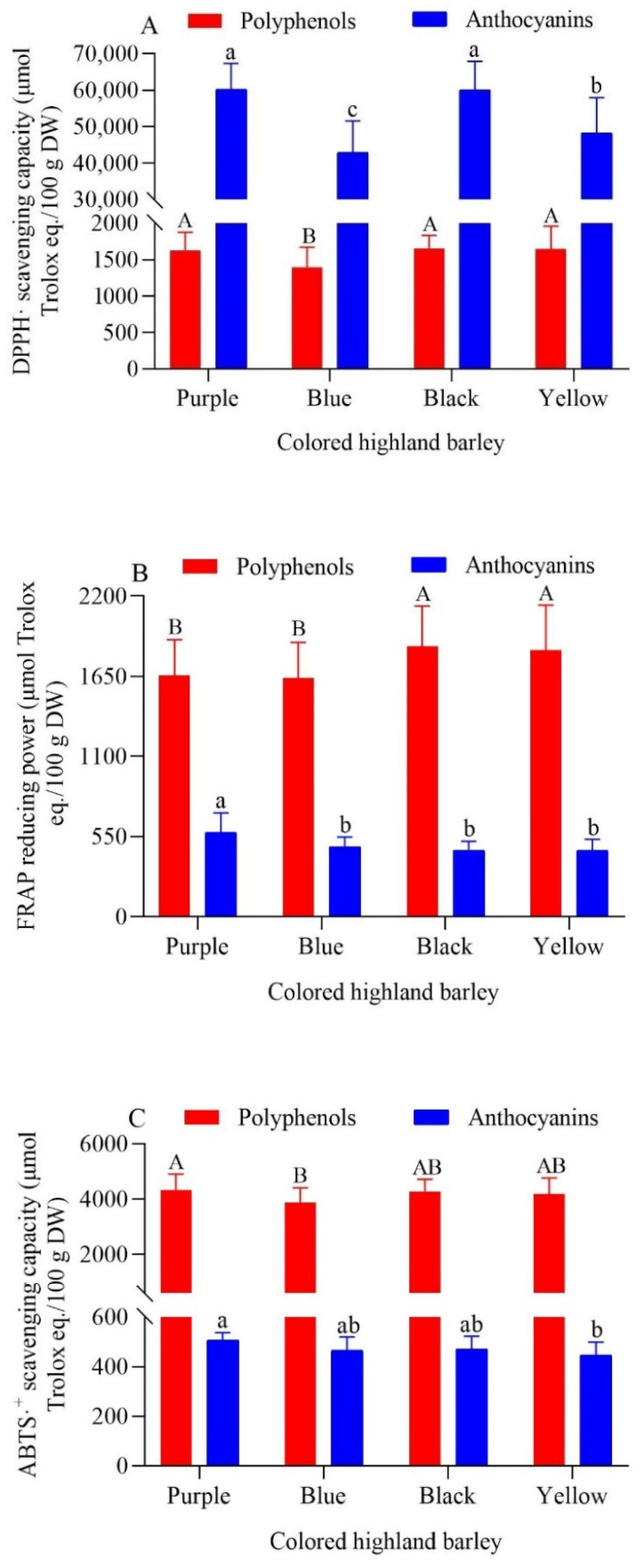
DPPH·-scavenging capacity (**A**), FRAP-reducing power (**B**) and ABTS·^+^-scavenging capacity (**C**) of polyphenols and anthocyanins from colored highland barley varieties. The uppercase and lowercase letters in the figure respectively indicate significant differences in the antioxidant activities of highland barley polyphenols and anthocyanins with different grain colors (*p* < 0.05).

**Figure 2 foods-11-02025-f002:**
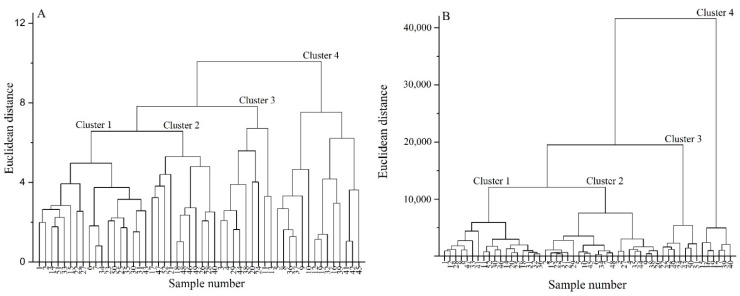
Cluster analysis results of highland barley varieties with different grain colors according to nutritional quality (**A**) and functional quality (**B**).

**Table 1 foods-11-02025-t001:** Highland barley varieties.

Number	Variety	Color	Number	Variety	Color
1	Z28	Purple	27	Heilaoya	Black
2	Z29	Purple	28	Kunlun17	Black
3	Z264	Purple	29	Kunlun 20	Black
4	Z299	Purple	30	949	Black
5	Z505	Purple	31	950	Black
6	Z510	Purple	32	Z523	Black
7	Z520	Purple	33	Z526	Black
8	Z525	Purple	34	Z528	Black
9	Z538	Purple	35	Z533	Black
10	14YN-748	Purple	36	Z536	Black
11	13Y11-5	Purple	37	Z541	Black
12	Ganziheiliuleng	Purple	38	Z560	Black
13	Yunqing 2	Purple	39	Beiqing 6	Yellow
14	14-Z530	Purple	40	Kunlun 12	Yellow
15	Beiqing 2	Blue	41	Kunlun 14	Yellow
16	Beiqing 4	Blue	42	Kunlun 15	Yellow
17	Beiqing 8	Blue	43	Kangqing 3	Yellow
18	Ganqing 4	Blue	44	Kangqing 6	Yellow
19	Beiqing 9	Blue	45	Ganqing 3	Yellow
20	Zangqing 320	Blue	46	Aqing 5	Yellow
21	Mengnong 1	Blue	47	Aqing 6	Yellow
22	Mengyuanlianglan	Blue	48	Chaiqing 1	Yellow
23	Xunhualianglan	Blue	49	Zangqing 25	Yellow
24	Walan	Blue	50	Duanbaiqingke	Yellow
25	Dulihuang	Blue	51	Changheiqingke	Yellow
26	Zangqing 690	Blue	52	Ganqing 5	Yellow

**Table 3 foods-11-02025-t003:** Analysis on total phenol, total flavonoid, and total anthocyanin content of colored highland barley varieties.

Phenolic Content (mg/100 g DW)		Purple	Blue	Black	Yellow
TPC	Mean	204.29 ± 23.78 ^b^	207.59 ± 19.65 ^b^	225.16 ± 27.12 ^a^	223.61 ± 34.17 ^a^
	Range	170.45 (No.5)–238.38 (No.12)	166.20 (No.25)–237.60 (No.23)	196.65 (No.38)–273.94 (No.28)	172.95 (No.47)–278.01 (No.50)
	CV/%	11.64	9.47	12.04	15.28
TFC	Mean	19.88 ± 4.65 ^b^	23.60 ± 2.19 ^a^	23.33 ± 3.20 ^a^	23.78 ± 4.43 ^a^
	Range	12.84 (No.8)–26.53 (No.13)	20.63 (No.16)–28.11 (No.26)	18.80 (No.35)–28.59 (No.29)	16.59 (No.47)–32.53 (No.50)
	CV/%	23.39	9.26	13.72	18.64
TANC	Mean	18.65 ± 11.44 ^a^	15.27 ± 4.62 ^b^	15.05 ± 4.76 ^b^	12.94 ± 3.70 ^c^
	Range	7.50 (No.10)–45.03 (No.13)	9.17 (No.20)–22.27 (No.15)	7.36 (No.31)–21.74 (No.36)	7.99 (No.42)–20.74 (No.39)
	CV/%	61.33	30.24	31.60	28.59

Note: Different letters after data within the same line indicate significant differences (*p* < 0.05). The numbers in parentheses are highland barley variety numbers. CV: coefficient of variation; TPC, TFC, and TANC: total phenol content, total flavonoid content, and total anthocyanin content detected by chemical method, respectively.

**Table 4 foods-11-02025-t004:** Profile of phenolic acids and flavonoids in colored highland barley grains (μg/g) (Mean ± SD).

	Compounds	Purple	Blue	Black	Yellow
Phenolic acids	Phloxol	2.88 ± 0.32 ^c^	3.71 ± 1.71 ^b^	82.05 ± 0.70 ^a^	1.50 ±0.17 ^d^
	Gallic acid	10.26 ± 0.12 ^b^	9.11 ± 0.29 ^c^	20.47 ± 0.92 ^a^	10.10 ± 0.26 ^b^
	Protocatechuic acid	36.63 ± 3.64 ^b^	24.98 ± 5.52 ^c^	22.93 ± 3.51 ^d^	40.88 ± 5.44 ^a^
	Chlorogenic acid	32.72 ± 0.98 ^c^	28.32 ± 1.08 ^d^	35.94 ± 1.43 ^b^	50.56 ± 0.84 ^a^
	2,4-Dihydroxybenzoic acid	2.52 ± 1.31 ^c^	6.53 ± 1.23 ^a^	0.68 ± 0.27 ^d^	4.86 ± 0.92 ^b^
	Vanillic acid	10.78 ± 4.35 ^b^	5.10 ± 0.84 ^c^	12.20 ± 0.90 ^a^	5.75 ± 1.71 ^c^
	Syringic acid	5.07 ± 0.24 ^a^	3.44 ± 0.25 ^b^	3.29 ± 0.34 ^a,b^	4.56 ± 0.35 ^a^
	*p*-coumaric acid	6.98 ± 0.37 ^c^	9.25 ± 1.26 ^b^	11.76 ± 2.46 ^a^	4.31 ± 0.75 ^d^
	Ferulic acid	4.15 ± 0.16 ^b^	2.52 ± 0.17 ^c^	4.24 ± 0.18 ^b^	7.22 ± 1.18 ^a^
	Salicylic acid	8.04 ± 0.19 ^a,b^	8.34 ± 0.44 ^a^	7.65 ± 0.16 ^b^	7.68 ± 0.33 ^b^
	benzoic acid	42.73 ± 4.28 ^a^	2.78 ± 0.46 ^a^	15.56 ± 1.43 ^b^	3.16 ± 1.46 ^b^
	*O*-coumaric acid	9.49 ± 3.2 ^a^	2.36 ± 2.03 ^c^	6.95 ± 0.42 ^b^	9.29 ± 0.44 ^a^
	Veratric acid	16.30 ± 0.28 ^b^	9.00 ± 0.78 ^c^	20.36 ± 2.15 ^a^	5.11 ± 1.36 ^d^
	TPAH	188.55 ± 8.1 ^b^	115.44 ± 5.58 ^d^	244.08 ± 17.15 ^a^	154.97 ± 7.64 ^c^
Flavonoids	Catechin	33.65 ± 4.36 ^b^	33.09 ± 1.71 ^c^	41.44 ± 2.71 ^b^	46.94 ± 2.72 ^a^
	Naringin	6.63 ± 0.43 ^a^	2.37 ± 0.16 ^d^	4.06 ± 0.44 ^b^	3.05 ± 0.76 ^c^
	Hesperidin	3.47 ± 1.12 ^a^	2.01 ± 0.92 ^b^	3.59 ± 1.22 ^a^	nd
	Myricetin	0.47 ± 0.18 ^b^	nd	0.57 ± 0.26 ^b^	2.68 ± 1.45 ^a^
	Quercetin	7.69 ± 4.22 ^b^	nd	5.53 ± 0.93 ^c^	12.21 ± 0.24 ^a^
	Naringenin	9.08 ± 0.36 ^a^	8.25 ± 3.70 ^b^	7.46 ± 0.22 ^c^	5.32 ± 0.10 ^d^
	Kaempferol	13.36 ± 0.19 ^a^	8.60 ± 7.66 ^c^	14.05 ± 0.35 ^a^	9.58 ± 0.46 ^b^
	Rutin	8.46 ± 0.82 ^c^	11.01 ± 5.55 ^b^	9.15 ± 0.45 ^b,c^	10.32 ± 1.12 ^a^
	TFH	82.81 ± 4.13 ^b^	65.33 ± 0.12 ^c^	85.85 ± 2.11 ^b^	90.10 ± 2.38 ^a^
	TPH	271.36 ± 69.11 ^b^	180.77 ± 57.92 ^d^	329.93 ± 177.02 ^a^	245.07 ± 105.55 ^c^

Note: Different letters after data within the same line indicate significant differences (*p* < 0.05). nd: not detected; TPAH, TFH, and TPH: total phenolic acid content, total flavonoid content, and total phenol content detected by HPLC, respectively.

**Table 5 foods-11-02025-t005:** Profiles of anthocyanins in colored highland barley grains (μg/g) (Mean ± SD).

Phenolic Acids	Purple	Blue	Black	Yellow
Cyanidin	0.94 ± 0.64 ^b^	1.06 ± 0.81 ^a^	0.64 ± 0.18 ^c^	0.66 ± 0.23 ^c^
Cyanidin-3-glucoside	30.83 ± 43.86 ^a^	2.61 ± 3.11 ^c^	4.91 ± 3.51 ^b^	0.91 ± 0.05 ^d^
Delphinidin	1.66 ± 1.13 ^b^	1.92 ± 0.15 ^a^	1.25 ± 0.48 ^c^	1.06 ± 0.40 ^d^
Delphinidin-3-glucoside	9.06 ± 9.89 ^b^	10.94 ± 14.46 ^a^	7.37 ± 6.12 ^c^	0.56 ± 0.76 ^d^
Malvidin	nd	nd	nd	nd
Malvidin-3-glucoside	0.30 ± 0.19 ^b^	0.34 ± 0.32 ^a^	0.33 ± 0.19 ^a^	0.16 ± 0.18 ^c^
Pelargonidin	nd	nd	nd	nd
Pelargonidin-3-glucoside	0.85 ± 1.16 ^a^	0.14 ± 0.19 ^c^	0.21 ± 0.20 ^b^	0.08 ± 0.09 ^d^
Peonidin	0.10 ± 0.12 ^b^	0.12 ± 0.13 ^a^	0.08 ± 0.12 ^c^	0.12 ± 0.11 ^a^
Peonidin-3-glucoside	nd	nd	nd	nd
Petunidin	0.31 ± 0.06 ^b^	0.38 ± 0.22 ^a^	0.31 ± 0.09 ^b^	0.40 ± 0.09 ^a^
Petunidin-3-glucoside	0.88 ± 0.70 ^c^	1.76 ± 1.91 ^a^	1.04 ± 0.87 ^b^	0.32 ± 0.24 ^d^
TANH	44.92 ± 51.34 ^a^	19.28 ± 17.27 ^b^	16.14 ± 9.46 ^c^	4.28 ± 1.34 ^d^

Note: Different letters after data within the same line indicate significant differences (*p* < 0.05). nd: not detected; TANH: total anthocyanins content detected by HPLC-MS/MS.

**Table 6 foods-11-02025-t006:** Pearson correlation coefficients between nutritional components and phenolic compounds.

Groups	Ash	Fiber	Fat	Protein	Amylose	β-glucan
TPC		0.3285 *				
TFC					−0.3116 *	
TPAH				0.3488 *	0.2900 *	
TPH		0.2813 *		0.3028 *	0.2885 *	
TANH					0.3923 **	
Gallic acid				0.3146 *		
*p*-coumaric acid			0.3038 *			0.2846 *
Veratric acid		0.4170 **			0.3262 *	
Kaempferol				0.3258 *	0.3502 *	0.3248 *
Naringenin					0.3115*	
Rutin	0.4480 **					
Syringic acid				0.3470 *		
Naringin		0.3213 *			0.3608 **	
Hesperidin		0.3745 **				
Myricetin					−0.2848 *	
Cyanidin				−0.3253 *		
Cyanidin-3-glucoside					0.3750 **	
Delphinidin				−0.3383 *	0.3242 *	
Delphinidin-3-glucoside						0.2808 *
Malvidin-3-glucoside					0.3916 **	
Pelargonidin-3-glucoside					0.3054 *	
Peonidin				−0.3022 *		0.3166 *

Note: ** indicates significance *p* < 0.01, * indicates significance *p* < 0.05. TPC and TFC: total phenols content and total flavonoids content detected by chemical method, respectively. TPAH and TPH: total phenolic acids content and total phenols content detected by HPLC, respectively. TANH: total anthocyanins content detected by HPLC-MS/MS.

**Table 7 foods-11-02025-t007:** Pearson correlation coefficients between polyphenol content and antioxidant capacity.

Groups	DPPH	FRAP	ABTS
TPC	0.6220 **	0.8734 **	0.7399 **
TFC	0.3699 **	0.6245 **	0.3475 *
TPAH	0.3289 *		0.3080 *
TPH	0.3427 *	0.2804 *	0.3449 *
Protocatechuic acid	0.2816 *		
Chlorogenic acid		0.3479 *	
Benzoic acid			0.4029 **
Catechin	0.2860 *	0.3215 *	

Note: ** indicates significance *p* < 0.01, * indicates significance *p* < 0.05. TPC and TFC: total phenols content and total flavonoids content detected by chemical method, respectively. TPAH and TPH: total phenolic acids content and total phenols content detected by HPLC, respectively. DPPH: DPPH·-scavenging capacity; FRAP: FRAP antioxidant capacity; ABTS: ABTS·^+^-scavenging capacity.

**Table 8 foods-11-02025-t008:** Pearson correlation coefficients between anthocyanin content with antioxidant capacity.

Groups	DPPH	FRAP	ABTS
TANC	0.2466 *	0.5532 **	0.3464 *
TANH	0.4675 *	0.6218 **	0.3807 **
Cyanidin		0.3413 *	
Cyanidin-3-glucoside	0.4200 *	0.6296 **	0.3279 *
Delphinidin		0.4034 **	
Delphinidin-3-glucoside			0.3147 *
Malvidin-3-glucoside	0.3187 *		
Pelargonidin-3-glucoside	0.5742 *	0.5230 **	

Note: ** indicates significance *p* < 0.01, * indicates significance *p* < 0.05. TANC: total anthocyanins content detected by chemical method. TANH: total anthocyanins content detected by HPLC-MS/MS. DPPH: DPPH·-scavenging capacity; FRAP: FRAP antioxidant capacity; ABTS: ABTS·^+^-scavenging capacity.

## Data Availability

The data presented in this study are available on request from the corresponding author.

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
