# Peer review of "Evaluation of Nutritional Components, Phenolic Composition, and Antioxidant Capacity of Highland Barley with Different Grain Colors on the Qinghai Tibet Plateau"

_foods, 2022, doi:10.3390/foods11142025_

Round 1
Reviewer 1 Report
Authors Dang et al. investigated the nutritional value and bioactives of 52 highland barley cultivars of the Qinghai Tibet Plateau. The work is interesting and original. The manuscript is well organised and written. The only remark I have is the Table 6 with correlation coefficients (not analysis), which, although statistically significant, are weak. Therefore, I suggest to delete the table 6 and give the most important facts in short within text. The related conclusion should be also revised.
Author Response
Reviewer 1:
Authors Dang et al. investigated the nutritional value and bioactives of 52 highland barley cultivars of the Qinghai Tibet Plateau. The work is interesting and original. The manuscript is well organised and written.
Answer: We would like to thank the reviewer for their thoughtful review of our manuscript.
(1) The only remark I have is the Table 6 with correlation coefficients (not analysis), which, although statistically significant, are weak. Therefore, I suggest to delete the table 6 and give the most important facts in short within text. The related conclusion should be also revised.
Answer: Thank you very much for your careful review of the manuscript and the valuable suggestion. At present, some studies have shown that there is a correlation between the nutritional quality and functional quality of grains, and the functional properties of grains can be preliminarily evaluated by analyzing the content of main nutrient components [1-3]. The author believes that the results found in this paper also have certain practical significance for guiding the selection and utilization of highland barley varieties with different grain colors on the Qinghai-Tibet Plateau. Therefore, we think after consideration that retaining the Table 6 section may be more beneficial for the integrity of the analysis.
The reason for the small correlation coefficients between nutrient components and functional components may be that a relatively large sample size was used in the present study, and there are certain degree of differences among different varieties. As pointed out by respected expert, the manuscript should give the most important facts in short within text. Thus we remove the insignificant data in the all correlation coefficient tables (Table 6, 7 and 8), and improve the relevant titles, descriptions and conclusions, to make the manuscript as reasonable as possible. Please refer to the red marked parts in “Abstract”, “3.6 Correlation between nutritional quality and functional quality”, “3.7 Correlation between polyphenol content and antioxidant capacity” and “3.8 Correlation between anthocyanin content and antioxidant capacity” in revised manuscript for details.
[1] Suriano, S.; Iannucci, A.; Codianni, P.; Fares, C.; Russo, M.; Pecchioni, N.; Marciello, U.; Savino, M. (2018). Phenolic acids profile, nutritional and phytochemical compounds, antioxidant properties in colored barley grown in southern Italy. Food Res. Int. 2018, 113, 221-233. https://doi.org/10.1016/j.foodres.2018.06.072.
[2] Tong, L.T.; Liu, L.Y.; Zhong, K.; Wang, Y.; Guo, L.N.; Zhou, S.M. Effects of cultivar on phenolic content and antioxidant activity of naked oat in China. J. Integr. Agric. 2014, 13(8): 1809-1816. https://doi.org/10.1016/S2095-3119(13)60626-7.
[3] Memon, A.A.; Mahar, I.; Memon, R.; Soomro, S.; Luthria, D.L. Impact of flour particle size on nutrient and phenolic acid composition of commercial wheat varieties. J. Food Compos. Anal. 2019, 86, 103358. https://doi.org/10.1016/j.jfca.2019.103358
Reviewer 2 Report
Overall, the job is very interesting. Although it must be said that the amount of the study material is so large that it could be used to create two separate works (e.g. on the nutritional value and the antioxidant values). Regardless, I want to say that the work is clear and understandable.
In the title, the term "comparative" could be omitted.
It would also be worth improving the Abstract, which in its current form is a set of sentences with too many digital results. In my opinion, a better solution would be to use in the Abstract most important statments from chapter "Conclusion".
Author Response
Reviewer 2:
Overall, the job is very interesting. Although it must be said that the amount of the study material is so large that it could be used to create two separate works (e.g. on the nutritional value and the antioxidant values). Regardless, I want to say that the work is clear and understandable.
Answer: Thank you very much for your careful review and constructive suggestions with regard to our manuscript.
- In the title, the term "comparative" could be omitted.
Answer: We agree with the reviewer and have removed the term “comparative” from the title of manuscript.
(2) It would also be worth improving the Abstract, which in its current form is a set of sentences with too many digital results. In my opinion, a better solution would be to use in the Abstract most important statments from chapter "Conclusion".
Answer: Thank you for your valuable comments and guidance. We have carefully checked the “Abstract” section and re-condensed the languages based on the main conclusions of different chapters to enhance the rationale and readability of the abstract as much as possible. Please refer to the red marked content in the abstract part of the revised manuscript for details.
Reviewer 3 Report
The manuscript entitle "Comparative Evaluation of Nutritional Components, Phenolic Composition and Antioxidant Activity of Highland Barley with Different Grain Colors on the Qinghai Tibet Plateau" has good scientific sound, however, the manuscript have several English, methods, and descriptive issues. The MS is unacceptable in its current form and need major revisions. The details for authors are as follows:
-Please rewrite the following sentences. These sentences are too long and different to understand. Additionally, in the abstract some sentences are too descriptive such as "Protocatechuic acid, catechin, cyanidin-3-glucoside, malvidin-3-glucoside and pel- 32 argonidin-3-glucoside were the main contributors to the scavenging of DPPH·, while chlorogenic 33 acid, catechin, cyanidin-3-glucoside, delphinidin and pelargonidin-3-glucoside were the main 34 contributors to the reduction of ferric iron, and benzoic acid, cyanidin-3-glucoside and del- 35 phinidin-3-glucoside were the main contributors to the scavenging of ABTS·+". Please re write and short this sentence, same is the case in conclusion.
-In the introduction section this sentence is too long. Have more than 8 lines of one sentences! It's better to rewrite and split into different part for better understanding. " In view of 104 this, this study selected 52 varieties of highland barley with different grain colors on the 105 Qinghai Tibet Plateau, compared and analyzed the nutritional components, phenolic 106 content and composition and antioxidant activity of highland barley with different grain 107 colors from the perspective of group, determined the quality characteristics and differ- 108 ences of highland barley with different grain colors, clarified the relationship between 109 main nutrients and phenolic content, and the impact of main phenolic compounds on 110 antioxidant activity, and scientifically evaluated the quality characteristics of highland 111 barley with different grain colors on the Qinghai Tibet Plateau".
-Please re-check the overall scientific English writing of the whole manuscript.
-In purple color the total anthocyanin contents is mentioned by author is "44.92±51.34a" in table 5. And the antioxidant activity in Figure 1, the author mentioned that DPPH activity is more than 60,000 mg Trolox/ 100g of DW. 60,000 mg= 60 grams. Which means 60 grams Trolox/ 100g of DW. Is it Possible? Is is too high?? It seems to be some major mistake in the protocol or in the description of the units. In the material and methods section the authors mention in 2.10. the unit as (µmol Trolox/100g DW) but in the figure 1 it is mentioned as (mg Trolox/ 100g of DW). Similar mistakes have been made in Figure 1B and C.
Additionally, everywhere the author mentioned DPPH radical scavenging activity whereas in figure 1 description it is written as ability. Please use uniform words. Interestingly, the DPPH activity formula is not mentioned in the 2.10., the author only mentioned this "(mg Trolox/ 100g of DW)" and form this standard we can only determine the DPPH capacity. DPPH activity and DPPH capacity is two different things. So, the author represent the DPPH capacity method and show the results of activity, and didn't written the formula of DPPH activity and also have dissimilar units.
Author Response
Reviewer 3:
The manuscript entitle "Comparative Evaluation of Nutritional Components, Phenolic Composition and Antioxidant Activity of Highland Barley with Different Grain Colors on the Qinghai Tibet Plateau" has good scientific sound, however, the manuscript have several English, methods, and descriptive issues. The MS is unacceptable in its current form and need major revisions.
Answer: Thank you very much for your careful review of the manuscript and your valuable comments. We have revised the content of the manuscript according to your comments, with a view to make the work more specific and clear.
(1) Please rewrite the following sentences. These sentences are too long and different to understand. Additionally, in the abstract some sentences are too descriptive such as "Protocatechuic acid, catechin, cyanidin-3-glucoside, malvidin-3-glucoside and pelargonidin-3-glucoside were the main contributors to the scavenging of DPPH·, while chlorogenic acid, catechin, cyanidin-3-glucoside, delphinidin and pelargonidin-3-glucoside were the main contributors to the reduction of ferric iron, and benzoic acid, cyanidin-3-glucoside and delphinidin-3-glucoside were the main contributors to the scavenging of ABTS·+". Please rewrite and short this sentence, same is the case in conclusion.
Answer: We are very sorry for the difficulty that our irregular writing has brought to your review. We have rewrite and short the sentences in abstract and conclusion. In addition, we also check the full text sentences to avoid such mistakes. Please refer to the red marked content in “Abstract” and “Conclusion” of the revised manuscript for details.
(2) In the introduction section this sentence is too long. Have more than 8 lines of one sentence! It's better to rewrite and split into different part for better understanding. " In view of this, this study selected varieties of highland barley with different grain colors on the Qinghai Tibet Plateau, compared and analyzed the nutritional components, phenolic content and composition and antioxidant activity of highland barley with different grain colors from the perspective of group, determined the quality characteristics and differences of highland barley with different grain colors, clarified the relationship between main nutrients and phenolic content, and the impact of main phenolic compounds on antioxidant activity, and scientifically evaluated the quality characteristics of highland barley with different grain colors on the Qinghai Tibet Plateau".
Answer: We are very sorry for our irregular writing of this sentence in introduction. We have rewritten the sentence and split it into different part for better understanding and reading. Additionally, we try our best to check and revise the similar issues in manuscript. Please refer to the red marked content in “Introduction” of the revised manuscript for details.
(3) Please re-check the overall scientific English writing of the whole manuscript.
Answer: Tanks for your careful review. We have carefully checked the writing of the manuscript and improved some words, tenses and sentences etc. as much as possible. Meanwhile, we also commission a native speaker to polish the article, with a view to improve the overall language quality of the work. Please refer to the red marked content in the revised manuscript for details.
(4) In purple color the total anthocyanin contents is mentioned by author is "44.92±51.34a" in table 5. And the antioxidant activity in Figure 1, the author mentioned that DPPH activity is more than 60,000 mg Trolox/ 100g of DW. 60,000 mg= 60 grams. Which means 60 grams Trolox/ 100g of DW. Is it Possible? Is it too high?? It seems to be some major mistake in the protocol or in the description of the units. In the material and methods section the authors mention in 2.10. the unit as (µmol Trolox/100g DW) but in the figure 1 it is mentioned as (mg Trolox/ 100g of DW). Similar mistakes have been made in Figure 1B and C.
Answer: Thank you very much for your careful correction, we are very sorry for this obvious mistake. After careful inspection and verification by the authors, here is an error of unit labeling. The correct unit in Figure 1 is μmol Trolox eq./100g DW, not mg Trolox eq./100g DW. We have redrawn the Figure 1A, Figure 1B and Figure 1C. Please refer to the revised manuscript for details.
(5) Additionally, everywhere the author mentioned DPPH radical scavenging activity whereas in figure 1 description it is written as ability. Please use uniform words.
Answer: Thank you for your careful review. We have unified the relevant words in the full text as “antioxidant capacity”, thus to make the expression more reasonable and avoid misunderstanding. Please refer to the red marked content in the revised manuscript for details.
(6) Interestingly, the DPPH activity formula is not mentioned in the 2.10., the author only mentioned this "(mg Trolox/ 100g of DW)" and form this standard we can only determine the DPPH capacity. DPPH activity and DPPH capacity is two different things. So, the author represent the DPPH capacity method and show the results of activity, and didn't written the formula of DPPH activity and also have dissimilar units.
Answer: Thank you very much for your careful review and guidance. Antioxidant results in this work are obtained by using Trolox as standard through the standard curve, and the unit is expressed as μmol Trolox eq./100g DW. As expert pointed out, they belong to antioxidant capacity, not antioxidant activity. Thus we have used the term of “antioxidant capacity” and the unit of “μmol Trolox eq./100g DW” uniformly in the full revised text. In addition, we have improved the method description sections, supplemented the information of standard curve equation, calculation procedure and correct unit, etc. Please refer to the red marked content in the revised manuscript for details.